# Digital Health Literacy of People with Intellectual Disabilities: A Scoping Review to Map the Evidence

**DOI:** 10.3390/ijerph22111748

**Published:** 2025-11-19

**Authors:** Dirk Bruland, Daniel Geffroy, Änne-Dörte Latteck

**Affiliations:** Faculty of Health, Bielefeld University of Applied Sciences and Arts, Interaktion 1, 33619 Bielefeld, Germany

**Keywords:** digital health literacy, health literacy, eHealth literacy, intellectual disabilities, digitalization, digital health, scoping review, health equity, health disparities

## Abstract

Digital technologies are revolutionizing health systems worldwide. People with higher digital health literacy are better equipped to access reliable health information, utilize telehealth services, and effectively manage their health through applications. However, a notable digital divide exists for people with intellectual disabilities, and the digitization of healthcare can limit their health opportunities. This scoping review examines the current evidence on digital health literacy among people with intellectual disabilities, emphasizing specific challenges and the need for tailored adaptations. Eleven articles from ten databases were included in the review. Although digital health literacy is becoming increasingly important, it is rarely discussed for people with intellectual disabilities. The term “digital health literacy” is not used, with the exception of one article. However, the focus is mostly on applicability and often at the functional level. The findings underscore that people with intellectual disabilities are underrepresented in research studies and interventions related to digital health literacy. Additionally, the results indicate the lack of a theoretical population-specific framework that focuses on competencies and life experiences. Participation in the digital world is a human right (UN CRPD). Addressing the digital gap is crucial, as improving digital health literacy can lead to better health outcomes, equitable access to health services, and reduced health disparities among people with intellectual disabilities. Based on the results, research directions for developing a population-specific framework for this highly vulnerable group are discussed.

## 1. Introduction

With great opportunities to improve individual and public health, digital technologies are noticeably transforming healthcare and public health systems around the world [1]. The “… field of knowledge and practice associated with the development and use of digital technologies to improve health” [2] is defined as digital health. It includes, for example, artificial intelligence, health data, health information systems, infodemic, digital therapeutics, the Internet of Things, and telemedicine [2]. Digital health has multiple benefits, including higher efficiency, increased independence, and increased communication options for service users, while service users’ social networks can also be easily informed [2,3]. An essential prerequisite for a successful digital transformation is the ability of different population groups and healthcare professionals to successfully deal with the digitalization boost, as well as the associated change and shift of diverse information and processes into the digital space [4]. For instance, people with higher education and income, of younger age, and with fewer health issues benefit more from digital health services than others [5]. Therefore, health inequalities need to be addressed. This is particularly true for people with intellectual disabilities.

Intellectual disability is characterized by significant limitations in intellectual functioning and adaptive behavior that originate before the age of 22, which distinguishes it from other diseases or syndromes [6]. Approximately 2% of the world’s population has an intellectual disability. Despite falling prevalence, the overall number of people with intellectual disabilities is increasing worldwide [7]. These individuals constitute a highly vulnerable group in terms of health. They live, on average, 27 years less than the general population and experience health inequalities at a high level [8]. They also tend to experience age-related health issues, particularly chronic illnesses such as reduced walking ability and a particularly high morbidity rate at an earlier stage of life compared to the general population. Additionally, they often receive fewer diagnostic and treatment interventions, including preventive and screening services, and have limited access to health promotion and prevention programs. There is also a notable lack of awareness regarding healthy lifestyles within this population group, with issues like poor nutrition, low physical activity, and improper medication use being common concerns [9]. Health literacy is a crucial factor for promoting health and health equity [10,11]. Strengthening health literacy supports patient empowerment, as it leads to a greater understanding of health and health systems, and increases the competence to be involved in decision making [12]. Due to the specific needs of people with intellectual disabilities, a target-oriented health literacy concept has been discussed in the literature [9,13,14,15]. Although digitalization is becoming increasingly important in health issues, to the best of our knowledge, the digital health literacy (DHL) of people with intellectual disabilities has attracted little attention so far. Although a “digital divide” of people with intellectual disabilities is still evident [16,17], they are increasingly included in the digital world, and smartphone and tablet use is becoming commonplace [18]. It can be assumed that DHL contributes to better health and empowerment for people with intellectual disabilities. Additionally, it strengthens “health equity approaches and accessibility for people with disabilities to promote an inclusive digital society with enhanced digital health skills” [1].

DHL is seen as more than the combination of the concepts of digital literacy and health literacy. Rather, it is seen as a complex, multidisciplinary concept that requires its own framework [19]. There are different definitions, and these are evolving based on the digital transformation. Therefore, there is no fixed term for DHL [20]. Ban et al. [20] (p. 8) defined DHL as “the ability to convert knowledge into practical actions by processing, communicating and utilizing health information from digital platforms, while continually regulating the knowledge translation process aligned with one’s health goals. This empowers individuals to make informed judgements related to the health continuum, ultimately enhancing their overall quality of life throughout the lifespan” [20]. Research indicates that people with higher levels of DHL report fewer illnesses, are more able to self-manage and participate in medical decisions, show greater ability to follow preventive public health measures, and are better able to manage their own mental state. However, low DHL can increase the health gap [21]. Therefore, the World Health Organization emphasizes the importance of incorporating DHL into public health strategies [1]. In order to ensure that people with disabilities are not left behind, a more comprehensive and differentiated understanding of DHL is strongly required [22].

The project “Strengthening digital health literacy of people with intellectual disabilities” (DGeKo MmgB) aims to develop, test, and evaluate a specific theoretical framework for strengthening the DHL of people with intellectual disabilities in a science-based, participatory, and practical manner. As a starting point, the current evidence and gaps of DHL for people with intellectual disabilities are explored via this scoping review. DHL is shaped by situational, personal, and environmental factors [23], and these factors are examined from the included studies. The following research questions served as a guide for the scoping review:What specific challenges are mentioned for people with intellectual disabilities in relation to digital health literacy?What needs-oriented adaptations are proposed for digital health literacy concepts for people with intellectual disabilities?

The results are analyzed and discussed in terms of implications to develop a framework of DHL for people with intellectual disabilities.

## 2. Methods

### 2.1. Overall

Scoping reviews are conducted when an initial orientation on the state of the research literature on a given subject is required. They provide a systematic method for determining the type of research published in the field of interest and for identifying research gaps [24]. A scoping review was conducted to map out the existing literature on the topic of DHL for people with intellectual disabilities based on the methodological guidance of Arksey and O’Malley [24]. The following steps were undertaken: (1) formulating the research questions (see the Introduction), (2) identifying relevant studies, (3) selecting relevant studies, (4) charting the data, and (5) collating, summarizing, and reporting the results [24]. The review is reported according to the Preferred Reporting Items for Systematic Reviews and Meta-Analyses-Scoping Review Extension (PRISMA-ScR) [25]. The PRISMA-SCR checklist is available as Appendix A. A scoping review protocol had been published beforehand [26]. This scoping review had also been registered on the Open Science Framework (OSF) (registration number: DOI 10.17605/OSF.IO/x4h72-v1).

The preliminary search was a significant step in determining the course of action. Several databases were checked in advance using the search terms “digital health literacy” AND “people with intellectual disabilities” without any restrictions. The hit rates were very low (overall 20 articles, but not fitting). Since it could be assumed that more information on DHL should be available, a sensitive search was chosen. This means that a broad understanding of DHL was used in order to identify as much of the existing evidence as possible.

### 2.2. Data Sources

To retrieve a wide range of potentially relevant interdisciplinary papers, a sensitive approach was chosen. The search was performed between 5 January and 12 February 2024 across 10 databases and an online resource. Articles were included if written in English and German. Additional language restrictions were not applied. The following databases were searched with English search terms: CINAHL via EBSCOhost, Eric via EBSCOhost, LIVIVO—The Search Portal for Life Sciences, PsycINFO via EBSCOhost, Medline via PubMed, Web of Science, and Wiley Online Library—while the following databases were searched with German search terms: CareLit, Fachportal Pädagogik/FIS Bildung, REHADAT, and Psyndex. The search strategies for each database were designed by D.B. and discussed within the team. The final strategy can be found in the Appendix A. As an additional search strategy, the reference list of the included articles was screened for relevant literature by two researchers (D.B. and D.G.). Grey literature is an extensive source of information. However, a grey literature search was not conducted for this scoping review. Google Scholar is regarded as a supplementary search, as it does not fulfill the transparency and completeness of a scientific information offer [27]. After the search process using academic databases, Google Scholar was searched on 27 April 2024 for additional literature. This search process was conducted by D.B. and yielded 1210 hits, but none of these articles were deemed eligible for inclusion.

### 2.3. Eligibility Criteria

The following criteria for inclusion were set:(1)Adults with intellectual disabilities in any living context, e.g., outpatient or inpatient care;(2)DHL is addressed;(3)Written in English or German;(4)All study designs, regardless of their design and quality, as well as grey literature, including dissertations and editorial articles;(5)Timespan from 2009 to 2023.

The exclusion criteria included the use of digital technologies (for health) without addressing DHL, e.g., measuring health outcomes in interventions. In addition, conference abstracts with no detailed information were also excluded.

### 2.4. Study Selection

The study selection process was performed by Rayyan [28], considering the verdict of two researchers (D.B. and D.G.). First, duplicates were detected and removed. Two reviewers (D.B. and D.G.) independently screened titles and abstracts and, subsequently, read the full texts of the papers that passed the initial screening. As mentioned above, the search terms “digital health literacy” AND “intellectual disabilities” were tested in the databases. The hit rate of articles was very low. Convinced that more information must be available, the search was expanded to include studies with tangential relevance. This means that articles that did not mention the words “digital health literacy” or “eHealth literacy” but had relevant information on this topic were included. For study selection, the authors agreed on building an understanding of adjacent concepts (see Table 1) and how they are distinct from DHL (see below).

Distinct from DHL, health literacy focuses on health information but ignores the use of digital technologies. For example, it is important to critically evaluate how health information arises on the Internet. This varies from other health information, like leaflets.

Digital literacy and media literacy broadly cover general digital information and skills applicable across many domains. DHL specifically pertains to the skills necessary to understand and utilize digital health information and tools for health-related purposes. Health information has greater consequences for life. It can directly impact well-being; incorrect or misunderstood health advice may lead to worsening of conditions or harm. Digital health information, in contrast to other information, must be transferred into personal health decisions relating to one’s own lifestyle. This transfer is more complex than, e.g., information to fix a smartphone. For this, factors such as personal characteristics like age, environmental factors, and especially individual risk factors have to be taken into account.

An article was included if it

Mentioned specific content of digital health information of people with intellectual disabilities;Included an application to improve access to health information or health management;Deepened the understanding of user experiences and identified support needs within social environments to access digital health information and transfer them into real life;Described learning approaches that mentioned digital health information;An assessment of risks and facilitators.

In cases of unequivocal voting for inclusion, the disagreement was resolved by discussion or, in the case of persisting disagreement, by a third researcher (Ä.D.L.).

### 2.5. Data Charting

Data charting forms were developed in order to answer the research questions (D.B., D.G., and Ä.D.L.). First, the characteristics of the included studies were described in an overview table, including author(s) and year of publication, country, challenges addressed and objective(s), sample, setting, theoretical foundation, and how the study addresses DHL (see Appendix A). This information was necessary to be transparent about which studies met the above-mentioned inclusion criteria. The complexity of DHL should be addressed by recognizing various dimensions that together play an important role in its shaping. To map the current knowledge, as a second step, categories were applied, taking personal (sociodemographics and individual skills), situational (current health concerns), and environmental (digital and social environment) factors into account [23] (Appendix A). Finally, the data were charted, focusing on the sub-questions, the main results of each article, specific challenges, and needs-orientated adaptions (Appendix A), which served as the basis for categorizing the results thematically via a qualitative content analysis [33]. 

## 3. Results

After starting with a general overview, situational, personal, and environmental factors are shortly illustrated. Following this, the sub-questions regarding specific challenges and needs-oriented adaptations are presented. This forms the basis for discussion of gaps and the need for a theoretical framework for strengthening the digital health literacy of people with intellectual disabilities. Finally, the results are summarized.

### 3.1. General Results

#### 3.1.1. Literature Search

The initial database search yielded 12,628 articles. After deduplication, we screened the titles and abstracts of 11,193 references; this was followed by full-text screening of 98 references. Finally, 11 articles were included for synthesis. The additional search strategies (searching Google Scholar and the reference lists of included studies) did not yield any additional relevant studies. The database search and study selection processes are illustrated in Figure 1.

#### 3.1.2. Study Characteristics

The publication dates ranged from 2011 to 2023. Approximately one-third (n = 4) of the 11 studies were published within the last two years of the search timeframe [34,35,36,37]. Seven studies used a qualitative research design [3,34,35], with four studies combining such design with an intervention [36,37,38,39], two studies used a quantitative research design [40,41], one was a literature review [42], and one study used a theoretical approach for designing an eLearning platform [43]. The most prominent study country was the UK [38,41,42]. Overall, only one of the included articles used the term digital health literacy, but without providing further explanation [34]. Two studies used the term health literacy [35,36]. These three studies were all published within the last year of the search timeframe. Other theoretical foundations could not be found, with one exception; Arachchi et al. [43] used learning theories as a foundation for designing an eLearning platform for health information. An overview of the included studies is provided in the Appendix A.

#### 3.1.3. Quality Appraisal

Appropriate instruments from the Joanna Briggs Institute (JBI) were used to assess the methodological quality and risk of bias of the included studies. The JBI Qualitative Research Checklist [44] was used for qualitative studies, while the JBI Checklist for Case Reports [45] was used for case reports, and the revised tool according to Barker et al. [46] was used for quasi-experimental studies. The JBI tool for analytical cross-sectional studies [45] was used for corresponding studies. Systematic reviews were assessed using the JBI Critical Appraisal Checklist for Systematic Reviews and Research Syntheses [47].

Overall, most of the included studies showed high to very high methodological quality with low risk of bias. This applies in particular to the study by Frielink et al. [3]. Refs. [37,38,39] were credible in key aspects of qualitative research, such as data collection, transparency of analysis, and traceability of results. However, a recurring point of criticism was the lack of explicit reflection by the researchers, which slightly increased the risk of interpretative bias. Some studies had limitations in their sample selection, for example, by recruiting participants through specialized institutions where people were particularly open to the respective research topic, such as [37,38]. This could have led to a positive selection bias. The study by Dam et al. [36] was assessed as being of limited quality overall; although the results provide added value to the review in terms of content, the description of the data analysis and the transparency of the presentation of results were particularly inadequate. The study by Salmerón et al. [40] was of low quality due to methodological ambiguities and several unaddressed bias risks. Chadwick et al. [41] also provided evidence of possible confounding factors, but these were not systematically identified as such or taken into account analytically. Significant methodological shortcomings were evident in the review by Sheehan and Hassiotis [42], which was rated as low in quality and high in risk of bias due to the lack of methodological information, for example, on study selection or bias minimization. It was not possible to evaluate the elaborated conference paper by Arachchi et al. [43] because it is grey literature, and the necessary quality criteria could not be verified.

In summary, it can be concluded that the quality of the included literature is predominantly high, although individual studies have methodological weaknesses that should be taken into account when interpreting the results.

### 3.2. Factors of Digital Health Literacy

Understanding DHL does not mean focusing exclusively on individual skills but rather encompasses a holistic approach. In this approach, individuals must be viewed in terms of their specific social context [23]. As such, attention to the interplay of different factors (situational, personal, and environmental) is required [23].

#### 3.2.1. Situational Factors

Situational factors include current acute and chronic health concerns. In included studies they were mostly discussed as a general health topic, like healthcare information [3,34,38,39,41,42,43] or general misinformation [40]. Savage [37] set examined the topic of health nutrition. Dam et al. [36] conducted two focus groups, each with ten people with mild to moderate intellectual disabilities. Four main topics were addressed: nutrition, medical specialization, patient information, and first aid. In line with that, Kuruppu Arachchi et al. [35] conducted structured interviews with people with intellectual disabilities to ask about their relevant health topics. It was found that most of the comments were about diet. However, topics were often determined through discussion with their doctors and family members or via advertising [35].

#### 3.2.2. Personal Factors

Sociodemographics, if any [41], were mostly provided as part of sample description [35,37,38,39,40,43]; the same was observed for individual skills, if listed [3,34,35,36,39,41,42,43]. Hall et al. [38] stated that skill levels varied regarding technology use (e.g., keyboard skills). In summary, if mentioned, individual factors were made in general statements in relation to the intellectual disability or understanding written information [36,40,42], age difference in using technology [3,42], and differences in gender, language, and education [34]. However, these factors were not included as part of any investigation, and detailed statements could not be found.

#### 3.2.3. Environmental Factors

The social environment was often described with a focus on support—mainly on the type of support that was needed [37,39] or how support was offered [38,40]. Some studies also mentioned that supporters have an important impact on the technology use of people with intellectual disabilities [41,42]. Frielink et al. [3] highlighted the importance of considering person-oriented and individual needs in general. Kuruppu Arachchi et al. [35] stated that support from staff is expected from people with intellectual disabilities. With regard to the media environment (like usability of digital tools), the included studies had focused on various aspects, which will be described below.

### 3.3. Challenges

#### 3.3.1. Health Information

The importance of health information was mentioned by all authors. The challenges for people with intellectual disabilities were often discussed in relation to (a) the characterization of intellectual disabilities and (b) the fact that health information does not meet the needs of people with intellectual disabilities [35,38,42].

(a) One study stated that “Within the social domain of adaptive behavior, a high percentage of people with intellectual disabilities are characterized in terms of ingenuity, credulity, and gullibility” [40]. It was indicated that, in terms of evaluating health recommendations, people with intellectual disabilities not just experience a delay in their development, but rather they have undergone atypical development [40]. For example, they relied mostly on their prior knowledge or referred more often to other sources for support, like teachers or parents, instead of integrating their knowledge with newly acquired information [40].

(b) Health information is typically presented in media in high complexity [36]. Moreover, online health information is often unregulated and may be of poor quality [42]. For example, Internet forums often contain misinformation and inadequate advice [40]. This is relevant to anyone who seeks information online, but people with intellectual disabilities are more vulnerable to such misinformation.

The problems of misinformation or less understanding of health information could result in potentially serious consequences [40]. Crucially, “… people with intellectual disabilities may have difficulties in taking in and retaining information in order to make … [own] decision, and therefore they may not get the treatment they need, or they may get treatment they didn’t want. Even when they agree to treatment, if they do not fully understand what is going to happen to them, they may refuse to cooperate. This could be distressing for the person, his or her caregiver, and the health care staff …” [38]. This has a negative effect on treatment success. Longer treatment may be necessary, like longer stays in hospital [38].

Research gaps were mentioned in the included studies. Minimal information is known about the perspective of people with intellectual disabilities on their health information needs and interests [35], or how people with intellectual disabilities interact with critical information [35]. The causes for the atypical evaluation of health information could only be speculated, which were suggested in one study to be related the previously mentioned adaptive behavior or a learned understanding of the importance of evidence to support facts. However, the authors stated that this conclusion needs to be considered with caution because it has not been verified [40].

#### 3.3.2. Applicability

Digital health environments are usually designed for the general population. This has an impact on access for people with intellectual disabilities. Applicability deficits exist due to the limited cognitive, linguistic, and functional abilities, as well as physical limitations [40,43].

This was demonstrated most clearly in the study by Sheehan and Hassiotis [42]. People with cognitive and linguistic limitations may struggle to access complex, text-heavy resources. Physical limitations can include limited fine and gross motor skills, as well as limited ability to use keyboards and computer mice. Sensory impairments are common and can make it difficult to view information on small mobile device screens [42]. Due to these limitations, more time is necessary to learn skills for using a new technology [3,42]. Indeed, unsuccessful prior experiences can discourage users with intellectual disabilities. Software updates can also present challenges regarding new navigation and interfaces [42].

One challenge in designing websites or health applications is to minimize the cognitive capacity required by people with intellectual disabilities to interact with the system and its content in order to maximize the cognitive resources available for the learning process [43]. The principles of universal design were discussed in the included studies as well. Universal design is a well-established concept and applies equally to electronic devices in general. Due to the cognitive, linguistic, and functional limitations, people with intellectual disabilities are different in terms of their technological capabilities, and they have very different requirements in the design of websites or health applications. Even websites created specifically for people with intellectual disabilities have been found to apply the principles of universal design inconsistently [42].

In general, people with intellectual disabilities are often not involved in the design of digital applications for health [43], and this topic is unexplored for this population group [3].

#### 3.3.3. Social Support

People with intellectual disabilities require social support even when interacting websites with universal design [3,37,40]. The level and type of support depend on the individual resources of each person. However, even people with more severe and profound intellectual disabilities, often without verbal communication, may still make use of limited computerized interventions with appropriate support [42]. However, the level and type of support needed was not described in detail in the included articles with such an intervention [36,37,38,39].

Several studies mentioned that the support provided depends highly on the attitudes and behaviors of supporters [41,42]. Supporters like family members or carers can encourage or discourage technology use [42]. For example, pessimistic views and high protection can lead supporters to discourage technology use [40,41]. Chadwick et al. [41] found that regularity of contact with people with intellectual disabilities had no significant effect on the perceived risks or benefits of being online for people with intellectual disabilities. However, having children in the household positively correlated with it. These authors identified a research gap, stating that more research and practice work needs to be performed to reduce misconceptions and counteract prejudicial assumptions of reduced ability and the need for childlike protection for online safety among people with intellectual disabilities [41]. Supporters’ expectations and perceptions about the use of digital health applications in support of daily functioning are also mostly unknown [3].

Other forms of support, such as face-to-face support, also need more research. For example, it is stated that “future research should investigate providing addition supports in similar situations such as a support coach or a peer group who are working towards similar goals” [37] (p. 331), or “blended support as well, as it remains unclear whether participants were unfamiliar with the term or also with the concept” [3] (p. 122).

### 3.4. Needs-Oriented Adaptations

#### 3.4.1. Applicability

As mentioned above, one challenge in designing websites or health applications is to minimize the cognitive capacity required by people with intellectual disabilities to interact with the system and its content in order to maximize the cognitive resources available for the learning process [43]. There was a high focus on applicability in the included studies [3,36,38,39,42,43].

One described method is to present complex information in an easy-to-read format supported by pictorial content [36]. Various guidelines and suggestions for easy-to-read language have been proposed for writing texts for people with intellectual disabilities. The main rules include using simple words and short sentences, providing examples to explain concepts, writing numbers as digits, and using sans serif fonts [36].

People with intellectual disabilities learn better from multimodal information sources [38]. In one study, content was delivered next to Easy Read English with an additional option for listening to an audio version of the text (e.g., screen reader), and videos were also used to present content [39].

Universal design was often advocated [3,36,38,39,42,43]. An overview of examples of the principles of universal design is presented in Figure 2.

In addition, the interface should appropriately use known symbols, logical arrangement with a course map, and less than a three-level structure to enhance navigability [43]. Dam et al. [36] (p. 3) reported that “The focus was on a visually reduced design from wordpress.com, which was adapted by the authors for this research. It focused on easy webpage navigation and navigation symbols familiar the participants such as the home button, the search symbol as well as the triple A for changing font sizes”.

It is essential to take a learner-centered perspective into account [43]. Development of a digital environment should integrate the needs, interests, existing knowledge, and capabilities of people with intellectual disabilities. For this, incorporation of theories like Social Cognitive Theory [43] can be beneficial. Co-creation should not simply be mentioned but implemented. For example, a facilitating factor is to include all stakeholders from the start [3]. Evaluation of products should include an assessment of accessibility and usability by people with intellectual disabilities [42]. Watfern [39], for example, invited experts who worked with people with intellectual disabilities to develop a health information website. At the final stage of the development process, a working prototype of the website was shown to two people with intellectual disabilities and their support workers for feedback as they worked through the activities. All feedback was incorporated into the development and refinement of the interface [39].

#### 3.4.2. Functional IT

Although rarely mentioned in connection with IT applications, functional IT is an important issue. The functionality of IT applications should be guaranteed, along with a lack of distractions; for example, pop-ups should be blocked [3,43].

#### 3.4.3. Support

It is noticeable that in the literature, supporters are broadly positioned with respect to involvement, attitudes, and an active exchange, but no further detail is provided [3,37,41]. The benefits of support were stated. For example, users who engage closely with support workers tend to be more engaged [39]. In other words, “… several authors have demonstrated that with appropriate training and support most people with a mild–moderate degree of intellectual disability can learn and retain basic computer skills” [42].

Four articles described intervention, which could consist of realized support for users with intellectual disabilities. Savage [37] recruited participants via social media for an intervention that promoted digital skills and health conditions. Hall et al. [38] recruited participants from a social center for an intervention that provided a well-resourced information technology suite. In total, 16 of the 20 participants had experience with technologies (yet eight did not often use a computer, and four did not use a computer at all). The reason for the low number of comments on support may be due to the characteristics of the selected group or the fact that support was provided on a situational basis [38]. Therefore, there is a research gap in this aspect.

#### 3.4.4. Training

Training has been considered to empower people with intellectual disabilities to use health information more independently. The included studies mentioned training on basic computer skills, familiarizing with supporting functions, and critically evaluating different information sources. Overall, the information level on this topic was minimal.

Arachichi [43] presented different studies on the basic computer skills required for using technology among people with intellectual disabilities. They stated that training should be twofold: “Basic training associated with improving basic computer skills among low-performance groups, and advanced training on essential functions, including ‘orientation and attention’ on the Internet” [43] (p. 16).

Familiarizing with supporting functions could be beneficial for some individuals, but they have to be trained, for example, on using the autocomplete function in the search box and suggested keywords at the bottom of the search results page, or applications of voice recognition for people who have low literacy skills.

Salmerón et al. [40] stated that people with intellectual disabilities need specific support to critically evaluate the credibility of different information sources. They claimed the importance of improving this ability. In our understanding, this could be enhanced via a training program [40].

Additionally, Arachichi [43] mentioned that training increases the ability of people with intellectual disabilities to navigate the Internet, provided that they receive continuous training. Continued training was not mentioned in any other articles [43].

#### 3.4.5. Manuals

Self-management manuals can be used to help understand health information. For this, a(n online) dictionary with simplified descriptions can be beneficial [35]. One study found that some participants can write health information and difficult-to-understand words in a book for future use [35], or they can print the information for use in real life [35]. Savage [37] used visual support for digital interventions. In a broader sense, these interventions can be used generally for description.

#### 3.4.6. Real-Life Relevance

Several studies stated that health information is highly relevant for the day-to-day healthy life experiences and health needs of people with intellectual disabilities [35,36,37]. Regarding publication year, this aspect was more frequently recognized in recent years. Arachchi et al. [43] stated that knowledge results from the combination of grasping and transforming experiences. Another study stated that “Specific (health) situations, social context and everyday life routines are important key factors for health literacy, which is considered as social practice” [34] (p. 2). As explained by Hall et al. [38], it is necessary to bridge the gap between information representation and experiential learning in a safe setting.

### 3.5. Summary 

A graphical overview was created to summarize the results at a glance (Figure 3). In this graphical overview, the categories and important points mentioned above are placed in relation to each other.

## 4. Discussion

Access to information and communication technologies, including the Internet, as well as access to understandable information, is stated as a human right (e.g., Article 9 of the United Nations Convention on the Rights of Persons with Disabilities). However, people with intellectual disabilities are underrepresented in health research and suffer from insufficient patient empowerment [28]. While research on digital health literacy (DHL) is progressing internationally, research on DHL in people with intellectual disabilities seems to be lacking. As digital transformations increasingly permeate the healthcare sector, it becomes imperative to address the digital divide, particularly for people with intellectual disabilities. The Global Strategy on Digital Health 2020–2025 [1] aims to promote an inclusive digital society by enhancing digital health skills like DHL as part of health equity initiatives and improving accessibility for people with disabilities. Addressing this is crucial because it can lead to better health outcomes, equitable access to health services, and reduced health disparities, ultimately benefiting community well-being as a whole and reducing health costs. It is recommended that future research endeavors focus on cultivating a more comprehensive and nuanced understanding of DHL. This enhanced understanding will facilitate the development of effective solutions that can address the specific needs of individuals and communities [22]. This scoping review was conducted to map out the existing literature and knowledge gaps regarding the DHL of people with intellectual disabilities.

### 4.1. Key Findings

Universally, the term DHL is used most frequently to describe the ability to find and use health information in order to address or solve a health problem using technology [21]. In contrast, in our review of studies focusing on adults with intellectual disabilities, only one article used the term DHL [34]. No other studies with terms like eHealth literacy or mHealth literacy could be included regarding people with intellectual disabilities. Although the use of health information to address or solve a health problem using technology was addressed, in the included studies, terms like health literacy [35,36] or digital literacy [36,40] were used, but not DHL. Since conducting the scoping review, we checked this point. Afterward, one article was found that used the term DHL, for caregivers of students under 18 with intellectual disabilities [48]. In addition, other similar terms also appeared, such as online health literacy [49]. We also used the AI tool “OpenEvidence” on 27 October 2025 with the prompt “Find articles addressing digital health literacy of people with intellectual disabilities.” Five articles were identified: two of these studies were already included [34,36], one was published after the search timeframe [49], and two studies were excluded for this scoping review for not addressing DHL [15,50]. For example, one study was a scoping review about an inclusive approach for design development, where the implementation phase remained underexplored [50]. In summary, DHL is not used as a standard definition for people with intellectual disabilities, and we can only speculate the reasons. For a long time, there has been criticism that the discourse around health literacy for people with intellectual disabilities hardly goes beyond the functional health literacy domain. Health-related information is mostly changed into seemingly easy-to-read language, and interventions are developed for time-limited and very specific settings, but few of these measures address the real needs of people with intellectual disabilities [13,14]. In all the included studies, it was mentioned that health information and applicability are not address the needs of people with intellectual disabilities. Although the focus is on access to health information or digital health applications for one’s own health, adaptation of technical matter most often focused on. Vázquez et al. [51] stated as well that most studies focused on the applicability of technology to improve digital performance in daily life, but little attention is paid to key points related to its use by people with intellectual disabilities for their own health or consideration of cognitive abilities. In this sense, the term “digital health literacy” is not used in research studies due to the functional understanding and focus on applicability. In our opinion, this means that the results cannot be clearly distinguished from other models, such as digital literacy. A common understanding of target-group-oriented DHL among people with intellectual disabilities is lacking, but this most essential for consistent research and application.

### 4.2. Current Use for a Population-Specific Theoretical Framework

Aside from evidence mapping, a question arises as to how the included studies have expanded or challenged existing models. As a reminder, DHL is seen as a complex, multidisciplinary concept that requires its own framework [19]. It is recommended that future research endeavors focus on cultivating a more comprehensive and nuanced understanding of the DHL of people with intellectual disabilities, highlighting the transactional model of Paige et al. [22,52]. A comprehensive understanding is required, encompassing not only the communication channel, the source of communicators, and the type of message, but also social practice in relation to social, relational, and cultural contexts [52]. This is in line with our search for factors influencing DHL [23]. Our review summarizes prior work and clearly highlights research gaps. Although there are good indications, the included articles did not allow for an in-depth analysis for a more comprehensive and nuanced understanding that extends or challenges existing DHL models to address the needs of people with intellectual disabilities. However, the summarized results with a focus on the adaptation side, in addition to co-creation and sustainability, are highly relevant for further theoretical and practical work (see Figure 3). Above that, an understanding of a fundamental approach is highly relevant for a population-specific DHL model.

### 4.3. Digital Health Literacy Approach

Other studies involving people with intellectual disabilities have also come to the conclusion that the focus is predominantly on the domains of accessing and understanding information, with fewer resources investigating the appraisal and application of digital health information [49,51]. Thus, as in the general health literacy debate of people with intellectual disabilities, there is still a need to go beyond a functional level [13,14] and the promotion of digital skills. A framework of DHL that acknowledges and is tailored to the specific needs of people with intellectual disabilities seems to be missing. Therefore, there is an urgent need for conceptual scientific debates on DHL that include people with intellectual disabilities, their experiences, perceptions, resources, and social contexts [9,14]. In our understanding, the concept of DHL focuses on competencies and resources that are important to make self-determined health-related decisions [9,14]. It is crucial to move beyond mere access to digital health information and equip individuals with the skills and knowledge to interpret, appraise, and utilize digital health information effectively in the context of their personal health needs and transfer such information into the social context of their lives, even if they are dependent on other people.

Regarding DHL, it is stated that people with intellectual disabilities always need support or are dependent on others [3,37,40]. Many individuals without disabilities choose others to make or influence decisions in their lives (e.g., doctors, financial planners, family members, and supervisors). It is important that individuals guide the process and act on own behalf in order to keep the greatest possible decision-making power [53]. This also counts for people with intellectual disabilities (UN Convention on the Rights of Persons with Disabilities). Enabling self-determined action, however, requires systematic planning for personalized, appropriate, and creative supports that leverage available and emerging resources to enable everyone to express preferences, make decisions, and identify valued outcomes [53]. For this, a framework is essential.

In addition to the mentioned social context, an important factor for self-determination is to challenge biases, including the ongoing impacts of deficit-based models of disability in existing policies and practices [53]. However, in developing a framework for DHL, the scoping review identified several knowledge gaps.

### 4.4. Knowledge Gaps and Future Directions

To develop a complex, multidisciplinary framework of DHL, three main research gaps need to be addressed: (a) real-life relevance, (b) assessment of digital health information, and (c) interplay of factors.

To the best of our understanding, the concept of DHL focuses on competencies and resources that are important to make self-determined health-related decisions. Instead of focusing on access to health information, the focus should be on people’s concerns and needs for health management and how to make health-related decisions. Access should be discussed specifically in relation to their relevant topics, needs, and skills. For this, real-life relevance is necessary to enable the participation of people with intellectual disabilities themselves. For accessibility, co-designing of websites and interventions has been mentioned [3,39,42]. Although some studies asked about favorite health topics [35,36,37], there is limited understanding of the health information interests of people with intellectual disabilities [43]. More research with a participatory approach is needed to analyze the health information interests of people with intellectual disabilities and how digital health information can be presented in an interesting way.

There is also a need to understand how people with intellectual disabilities engage with digital health information and apply it to their lives. Little is known about the learning needs [43] of people with intellectual disabilities and how they interact with critical information online [35]. An atypical development was reported to be the cause of the limited ability to evaluate health information [40]. This provides a basis for developing effective training and support programs, as well as supportive materials, and it is necessary to move beyond the functional level to address the complexity of DHL [19]. As an additional challenge to be addressed in promoting digital health, people with intellectual disabilities often encounter difficulties in understanding digital social situations (Netiquette) [54].

This scoping review pays attention to the interplay of factors of DHL (situational, personal, and environmental) [23]. People with intellectual disabilities share a diagnostic criterion (typically lower IQ tests and deficits in adaptive behavior) and experiences connecting to similar life trajectories, especially with regard to institutional involvement, but they are highly heterogeneous [55,56]. In the included studies, general statements were made regarding the use of digital technologies among people with intellectual disabilities. However, little attention was paid to the impact of sociodemographics or individual skills. Further research is needed to evaluate the impact of these factors and address the heterogeneity of the population with intellectual disabilities [15,36]. Analysis should be conducted on how situational factors, personal characteristics, and social environments, including social support, social practices, and institutional policies, interact. A holistic approach that considers these factors will enhance the personalization and effectiveness of DHL initiatives [23,53].

### 4.5. Limitations

DHL can be a somewhat confusing construct despite the current importance of the concept, with its definition depending on the researchers’ approach and perspective. When conducting this scoping review, a major challenge was that the term DHL or related terms did not appear in titles or abstracts, with one exception. All articles were screened for their relevance to the topic of DHL among people with intellectual disabilities. In the context of literature reviews, it is imperative that the inclusion and exclusion criteria are meticulously delineated in clearly defined categories, and a codebook for article assessment is developed. The classification of articles into specified categories is conducive to ensuring reproducibility. The initial objective of the present study was to integrate existing target-group-specific DHL models. However, since no population-specific model was found, our aim was modified to conduct a sensitive analysis to obtain as much information as possible about the digital health literacy of people with intellectual disabilities, considering articles with tangential relevance. During the review process, new information was found that prompted us to question the basic criteria, and the research team felt it was essential to rediscuss the inclusion and exclusion criteria. As a result, it seemed more appropriate for us to proceed with a sensitive analysis than to disregard any possible information, given the limited data available. During the search process, two researchers evaluated all articles independently. However, during the study selection process, the researchers regularly coordinated with each other to discuss the selection criteria without mentioning a specific article. This was necessary in the search for studies with tangential evidence in order to ensure a common understanding of DHL and conserve limited resources. Discrepancies in the inclusion of articles were discussed afterward. Despite further clarification of the DHL definition and inclusion criteria due to the inclusion of studies with tangential evidence, study selection could be subjective and may not allow for replicability. Other researchers could have made different decisions. For example, Chadwick et al. [41] and Salmerón et al. [40] mentioned health as a subordinate topic, but for our research team, health is highly relevant for understanding the DHL of people with intellectual disabilities. Therefore, reasons for including studies with tangential evidence were added in the overview (Appendix A) to ensure greater transparency about this subjective decision. Due to the regular consultations taken during the study selection process, we decided to not check for intra- and interrater reliability.

Only a few studies met the eligibility criteria. Moreover, the included studies were very heterogeneous, making it difficult to conduct a thematic analysis. Despite the challenges, the objective of the scoping review was fully achieved by exploring the current evidence of DHL among people with intellectual disabilities, addressing specific challenges and needs in promoting DHL in this population group, and identifying knowledge gaps.

Overall, the included studies were of predominantly high to very high quality with a low risk of bias, indicating a solid evidence base. Studies with clear data collection and transparent analysis were particularly convincing (e.g., [3,34,39]). However, recurring weaknesses included a lack of reflection on the part of the researchers and selection bias due to recruitment from specialized institutions (e.g., [37,38]). Some studies (e.g., [36,40]) also showed deficits in the transparency of data analysis and in dealing with potential confounding factors. Overall, the methodological quality of the included literature can be assessed as predominantly robust, although weaknesses should be taken into account when interpretating the results.

A grey literature search should be conducted with high standards in a transparent manner, which can be extensive. Therefore, the decision to not conduct a grey literature search was made due to resource limitations. Our focus was mainly on academic sources of literature, which runs the risk of omitting potentially relevant information. This limitation will be addressed at a later stage of the project.

Another main limitation is evident: The aim of our study was to find out about the DHL characteristics of people with intellectual disabilities. However, this turned out to be difficult because not all articles provided a definition for DHL, and this term was often overlapped with related constructs such as digital literacy. The review was nevertheless able to provide useful information for our aim of developing a population-specific theoretical framework.

### 4.6. Implications and Benefits for Developing a Population-Specific Theoretical Framework of Digital Health Literacy

Due to the limited evidence, a population-specific framework of DHL that accounts for the situational, personal, and environmental realities of people with intellectual disabilities cannot be developed. However, the results of this scoping review will be used to guide the process of the research project DGeKo MmgB:A resource-oriented approach is central. The focus will be on people with intellectual disabilities and their health needs, and how validated digital health information and digital health applications can be applied to promote health management will be analyzed. Personal skills and social support will be considered.Real-life relevance has to be researched. This will be conducted utilizing a participatory approach and proxy surveys. For example, both people with intellectual disabilities and their support persons will be surveyed about relevant health topics and current use of digital technologies for health information and health management. Perspectives on benefits and risks, as well as motivation to use digital technologies for health, will also be surveyed.Our network of practitioners will be consulted to incorporate non-academic knowledge base. This will ensure that important sources of information and practical knowledge will be included.An initial framework will be developed based on the research results obtained. A holistic intervention approach will be taken to develop the framework. The focus will be on digital health strategies to address the learning needs of people with intellectual disabilities and their critical interaction with digital content. The discussion groups should involve different stakeholders, including people with intellectual disabilities and their support persons, as well as institution managers.The implementation of the framework in practice will be analyzed, and the framework will be modified accordingly.

As an expected result, a clear definition of “digital health literacy” that acknowledges and is tailored to the specific needs of people with intellectual disabilities will be developed. The derived benefits will serve as a basis for developing tailored interventions to strengthen digital health literacy, aiming to achieve high acceptance and good outcomes. Through better self-management and engagement in health decisions, a higher quality of life and a better long-term disease control are expected [21].

## 5. Conclusions

Digital technologies can play an important role in contributing to better access to needs-oriented health information and may include assistive support functions. The lack of comprehensive knowledge about DHL among people with intellectual disabilities and the lack of target-oriented adaptation for this population group hinders the development of effective interventions. There is an urgent need for research on DHL that specifically examines how people with intellectual disabilities engage with health information and use digital applications for health management. The focus of research must be broader than the applicability of technology to improve performance and encompass the influence of online information and supporters in making decisions regarding one’s own health. Strategies to improve DHL should be designed, taking into account the perspectives of people with intellectual disabilities. From a public health perspective, addressing these research gaps is crucial, as improving DHL can lead to better health outcomes, equitable access to health services, and reduced health disparities, ultimately benefiting community well-being as a whole and for people with intellectual disabilities. The project DGeKo MmgB will conduct further research based on the results of this scoping review. In general, the DHL of people with intellectual disabilities must be given greater consideration to strengthen a person-centered conceptual framing. As in the general health literacy debate, the focus must be on promoting people’s ability to make their own health decisions.

## Figures and Tables

**Figure 1 ijerph-22-01748-f001:**
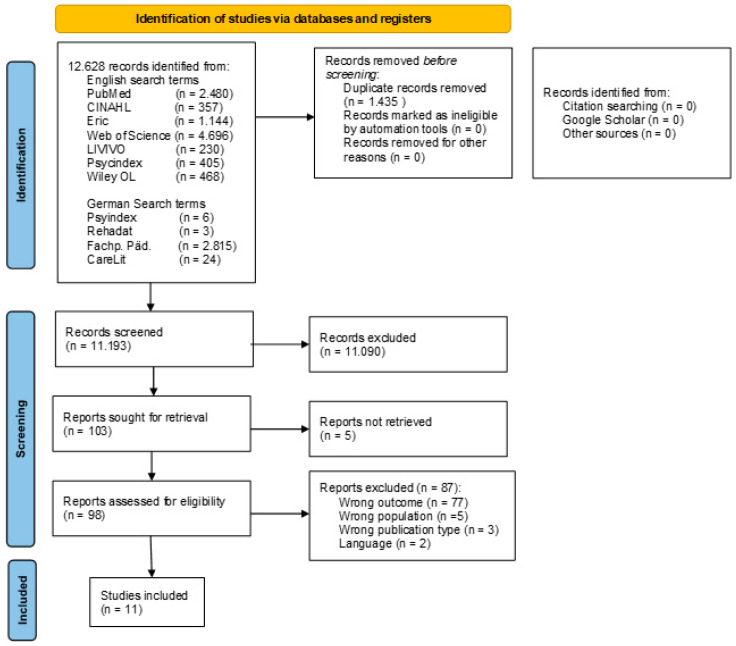
PRISMA flow diagram [25].

**Figure 2 ijerph-22-01748-f002:**
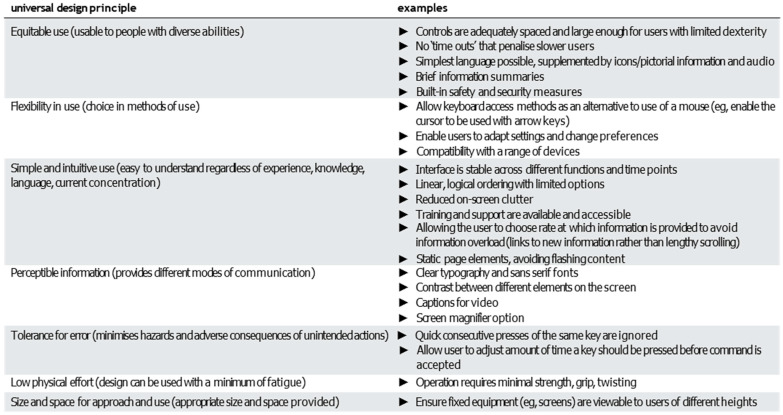
Principles of universal design can be used to optimize the accessibility of digital technologies [42].

**Figure 3 ijerph-22-01748-f003:**
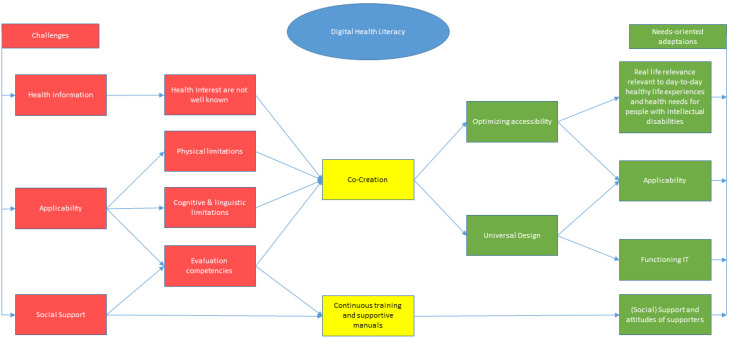
Graphical summary of the results.

**Table 1 ijerph-22-01748-t001:** Basic understanding of adjacent concepts.

Literacy	Definition/Description
Digital literacy	Digital literacy is “the ability to use information and communication technologies to find, evaluate, create, and communicate information, requiring both cognitive and technical skills” [29]. The Digital Competence Framework for Citizens (DigComp 2.1) contains five competency areas: (a) information and data literacy, (b) communication and collaboration, (c) digital content creation, (d) safety, and (e) problem solving [30].
Media literacy	Media literacy focuses on “use of critical thinking to parse or create mass media, especially as a consumer in an age of online misinformation and disinformation. … just as the latter refers to an ability to read, write, and understand words and phrases, the former refers to an ability to analyze, evaluate, and produce various kinds of media.” [31]
Health literacy	Health literacy is defined as the ability to obtain, read, understand, and use healthcare information to make appropriate/informed health decisions [32].
eHealth literacy and media health literacy	During the search, the terms eHealth literacy and digital health literacy were used interchangeably. We agree that digital health literacy is an expansion of eHealth literacy. eHealth literacy, as well as media health literacy—combining media literacy and health literacy—is seen as a key attribute of digital health literacy [20]. This decision has no effect on the results because both terms were used in the search strategy.

## Data Availability

The original contributions presented in this study are included in the article/Appendix A. Further inquiries can be directed to the corresponding author.

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
