# Peer review of "Digital Health Literacy of People with Intellectual Disabilities: A Scoping Review to Map the Evidence"

_ijerph, 2025, doi:10.3390/ijerph22111748_

Round 1

Reviewer 1 Report

Comments and Suggestions for Authors

This is a solid and timely scoping review of 11 studies on digital health literacy among people with intellectual disabilities. It identifies substantial gaps in the evidence and offers a constructive discussion of future research directions; it is also encouraging to see the topic explored with this level of depth.

The paper’s strengths include a clear introduction that establishes relevance and appropriate aims given the knowledge gap, a well-described method, and a discussion that distils the main points, highlights the gaps, and outlines sensible next steps.

The main area for improvement is the results section, which is heavy to follow and dominated by dense tables that function more as an information dump than a synthesis. Table 1 could be moved to an appendix with a concise summary in the main text; Tables 2 and 3 would benefit from clearer spacing between rows and reduced detail, or be shifted to supplementary material with a strengthened narrative synthesis in the results.

As an example, section ‘3.4.1. Challenges’ sub themes could have been explored in a bit more detail for the three categories of health information, applicability and social support. A separate section with narrative explanation could have been presented for this rather than all detail being presented in the Tables. And then table 4 wouldn’t be necessary as this is just a repeat of the information.

Minor points:

Minor grammar issues need to be improved in places, particularly in the discussion section (e.g. line 301 -302)

Minor issues: missing full stop line 166, use of a full stop rather than a comma in line 164 within a number.

Use of a citation number rather than a capital letter to begin a sentence line 191 (same again later in the results section)

Semi-colon rather than a colon was used line 193

Page numbers for quotes?

Typo line 271

Comments on the Quality of English Language

See earlier comments the paper will need editing for grammar

Author Response

All comments are in the attached file. 

Reviewer 2 Report

Comments and Suggestions for Authors

This manuscript addresses a highly relevant and underexplored topic: digital health literacy (DHL) among individuals with intellectual disabilities, employing a systematic and structured approach. The topic is timely, given the increasing reliance on digital health tools and platforms, and the manuscript’s emphasis on equity, accessibility, and inclusion is significant in public health and disability research. One of the strongest aspects of the paper is its recognition of a critical gap in the literature, as well as its effort to synthesize existing studies to map current knowledge and highlight research needs. The review also effectively identifies the roles of social support, environmental context, and real-world relevance as pivotal factors influencing digital engagement for this population. These elements position the paper as a meaningful contribution to the field. That said, several areas require strengthening to elevate the manuscript to the standards of a top-tier peer-reviewed journal. A key area in need of improvement is the conceptual framing of digital health literacy itself. Although the manuscript notes that only one of the included studies explicitly uses the term DHL, it does not fully interrogate why this terminology is absent from much of the literature or explore the implications of this gap. A more critical discussion is needed on how DHL differs from or overlaps with related constructs, such as health literacy, eHealth literacy, and digital literacy. Additionally, the paper would benefit from proposing a refined, population-specific definition of DHL that accounts for the cognitive, social, and support-related realities of individuals with intellectual disabilities. Without such conceptual clarity, the review risks remaining too descriptive rather than shaping future discourse.

The analysis of the literature also needs greater depth and critical interpretation. While the paper successfully identifies knowledge gaps, barriers, and support needs, these findings are often presented descriptively without sufficiently analyzing why these issues exist or what their broader implications are. For example, the observation that most interventions focus on technical usability rather than meaningful engagement is important; however, the discussion could be further expanded by examining underlying causes, such as a lack of participatory design practices, funding limitations, or systemic biases in digital health development. The manuscript should also expand on how atypical information-processing patterns and evaluation strategies among people with intellectual disabilities affect their interaction with online health content and how these insights could inform the design of future interventions. The presentation of results is another area that could be strengthened. While the inclusion of tables and thematic analyses is thorough, the level of detail can sometimes obscure key findings, making it difficult for readers to discern the broader patterns and implications. The results section would be more impactful if organized around central thematic categories, such as barriers, facilitators, support structures, and design considerations, and if each theme were illustrated with clear examples. Additionally, integrating findings more explicitly with the discussion could help the narrative flow more coherently and highlight how each result contributes to the overall understanding of DHL in this context.

The discussion should also move beyond identifying what is missing to offering more concrete, actionable recommendations for future research, policy, and practice. For instance, the paper could propose specific strategies for co-designing digital interventions with individuals with intellectual disabilities and their caregivers, outline approaches to embedding inclusive design principles into technology development, or suggest frameworks for incorporating digital literacy training into health and social care services. These recommendations would not only enhance the practical relevance of the review but also position it as a guiding resource for researchers and practitioners working at the intersection of digital health and intellectual disability.

From a methodological perspective, the review is sound; however, the authors could enhance the transparency of their approach by providing more detail on how the inclusion and exclusion criteria were applied and how studies with tangential relevance were evaluated. Although scoping reviews typically do not include formal quality appraisal, a brief discussion of the methodological limitations of the included studies would help contextualize the strength of the evidence base and clarify the reliability of the conclusions.

Finally, the language and presentation of the manuscript would benefit from careful revision to improve clarity, readability, and flow. Some sentences are lengthy and complex, which makes key arguments harder to follow. Simplifying sentence structure, improving transitions between ideas, and maintaining consistency in terminology (for example, using “intellectual disabilities” consistently throughout) would enhance the manuscript’s accessibility for an international readership.

In conclusion, this manuscript makes a valuable contribution by synthesizing the limited but growing body of research on digital health literacy among individuals with intellectual disabilities. With a more robust conceptual framework, deeper critical analysis of findings, clearer narrative structure, and more specific recommendations for future research and practice, the paper has the potential to make a significant impact in both academic scholarship and applied public health. Refining the language and improving the presentation will further enhance its clarity, making it a strong candidate for publication in a high-impact journal.

Comments on the Quality of English Language

The quality of English in this manuscript is overall understandable. It conveys the main ideas, but it does not consistently meet the standards expected of a top-tier peer-reviewed journal. While the vocabulary is appropriate and the key arguments are present, the writing often lacks the clarity, precision, and flow necessary to fully capture the complexity of the topic. Because the subject matter, digital health literacy, accessibility, and intellectual disability, is nuanced and interdisciplinary, precise and polished language is essential for readers to grasp the significance and implications of the work.

One of the manuscript's main strengths is that it attempts to tackle a complex topic comprehensively and employs appropriate academic language. However, there are several areas where the writing could be improved to strengthen readability and impact. Many sentences are overly long and packed with multiple ideas, which makes the text harder to follow. Breaking these into shorter, more direct sentences would immediately improve clarity. There are also occasional grammatical errors, awkward phrasing, and inconsistent verb tenses. For example, some sentences shift from past to present tense within the same thought, which interrupts the flow and can confuse the reader. Careful proofreading would strengthen the overall quality of the writing. More consistent use of verb tenses would also make the text clearer and easier to follow. Typically, the past tense should be used when describing study findings, while the present tense is more appropriate for general knowledge.

Word choice could also be more precise. Phrases like “some studies,” “various factors,” or “certain gaps” are too vague and weaken the argument. Rephrasing them with more specific language and examples would strengthen and enhance the analysis's impact. Additionally, some key terms are used inconsistently or without clear definitions. Defining them early and using them consistently throughout would make the paper more cohesive and easier to follow.

Organization and paragraph structure could also be tightened. In several sections, especially the results and discussion, multiple ideas are presented together without clear transitions, which affects the logical flow. Reorganizing paragraphs so that each focuses on a single main point, supported by evidence and linked clearly to the overall argument, would improve the readability and coherence of the manuscript. Finally, transitions between major sections could be smoother. At times, the paper jumps from results to implications without clearly connecting how one informs the other. Stronger linking sentences would help guide the reader and reinforce the paper’s main arguments.

In short, the English is serviceable, but revisions focused on simplifying sentence structure, tightening grammar, clarifying terminology, and improving transitions would significantly elevate the quality. These changes would not only make the manuscript easier to read but also allow the strength and importance of the research to come through more clearly.

Author Response

All comments are in the attached file. 

Reviewer 3 Report

Comments and Suggestions for Authors

Thank you for the opportunity to read this thoughtful and timely scoping review. The manuscript addresses an important gap in the literature by examining digital health literacy (DHL) among people with intellectual disabilities. This is a vulnerable and underrepresented population, and the review has the potential to make a valuable contribution to the field.

Overall, the paper is well structured and clearly motivated. I particularly appreciate the comprehensive search strategy and the emphasis on both challenges and adaptation needs. There are, however, several areas that would benefit from further refinement.

Major Comments

  1. Conceptual Framing
    The manuscript would benefit from a clearer conceptualization of DHL in relation to adjacent concepts (eHealth literacy, digital literacy, health literacy). A concise figure or conceptual table might help clarify the authors’ approach.

  2. Research Questions
    The research questions are currently embedded in the methods section. Stating them explicitly at the end of the introduction would sharpen the focus and help the reader follow the study’s logic.

  3. Methodological Transparency
    The search strategy is detailed, but please clarify:

    • Were additional language restrictions applied?

    • How was grey literature systematically approached?

    • Was interrater reliability assessed during the screening process?

  4. Analytical Depth
    The results section is rich but primarily descriptive. Consider adding a thematic figure or conceptual model to highlight key insights and relationships between challenges and adaptations.

  5. Critical Reflection
    Even without formal quality appraisal, some reflection on the methodological quality and limitations of included studies would strengthen the manuscript.

  6. Discussion Structure
    Consider restructuring the discussion to clearly separate key findings, implications for practice and policy, and research gaps. This will make the contribution more visible and reduce repetition.

______________________________________________________

Minor Comments

  • Proofread carefully for typos and grammatical issues (e.g., “litearcy” in the abstract).

  • Standardize citation formatting throughout.

  • Tables are informative but dense; consider improving readability.

  • Define abbreviations (e.g., DHL, ID) consistently on first use.

  • Expand the PRISMA diagram to include reasons for exclusion at the full-text stage.

    Overall Impression

    This is an important and much-needed review. With improvements in conceptual clarity, synthesis, and structure, the manuscript could make a meaningful contribution to research on digital health equity for people with intellectual disabilities. I encourage the authors to further develop their thoughtful work.

Comments on the Quality of English Language

Since I am not an English language specialist and English is not my native language, I do not consider myself fully competent to assess the linguistic dimension of the manuscript. I kindly ask that you involve a language expert for this purpose. I have pointed out the language errors that I was able to notice from my own perspective and level of language proficiency.

Author Response

All comments are in the attached file. 

Round 2

Reviewer 1 Report

Comments and Suggestions for Authors

The authors have addressed the comments from the first review. The paper needs to be checked for typos and issues with grammar.

Author Response

Comment: "The authors have addressed the comments from the first review. The paper needs to be checked for typos and issues with grammar."

Dear Reviewer,

we like to thank you again for your time and valuable feedback. The article benefited greatly from the feedback. We did not receive any feedback from you regarding the revision. One reviewer had additional comments. The manuscript has been resubmitted with minor additions to clarify certain passages, but does not contain any significant changes. We will perform English editing as soon as the article is accepted in terms of content. 

Reviewer 2 Report

Comments and Suggestions for Authors

The manuscript explores an important and relevant topic with clear potential to contribute to the field. Its strengths include a coherent structure, accessible writing, and an evident effort to link theory with practice. However, the paper does not yet meet the depth and rigor required for a top-tier journal. The literature review summarizes prior work but lacks synthesis and critical analysis, making it unclear how the study extends or challenges existing theories. The methodology also needs greater transparency, including detailed descriptions of sampling, instruments, and analytic procedures to ensure replicability.

The results are clearly presented but require stronger interpretation; some conclusions overreach the available data. The discussion section should be revised to connect findings back to the theoretical framework and to clarify both practical and theoretical implications. Key sections, particularly limitations and future research, are missing or underdeveloped. Overall, the paper shows promise but needs substantial revision. Strengthening the conceptual framing, expanding methodological detail, and deepening analytical interpretation will significantly enhance the manuscript’s clarity, rigor, and overall scholarly impact.

Comments on the Quality of English Language

The quality of English in the manuscript is generally clear and readable, but refinement is needed to meet the standards of a top-tier academic journal. The writing effectively conveys key ideas, yet it occasionally lacks precision and consistency in tone. Sentence structure is sometimes wordy or repetitive, which can obscure the main points. The author should aim for more concise phrasing and smoother transitions to enhance readability and flow. There are minor grammatical and punctuation errors throughout the text, as well as inconsistencies in tense and formatting of citations. Attention to subject-verb agreement, article usage, and proper academic phrasing will strengthen the manuscript’s professionalism. Additionally, several sentences could be restructured to improve clarity, especially in sections where complex ideas are presented.

Overall, the English language quality is adequate for understanding the content, but requires careful editing and proofreading to achieve academic polish. A thorough language review, preferably by a native or professional academic editor, would help ensure the manuscript meets the stylistic and linguistic expectations of high-impact journals.

Reviewer 3 Report

Comments and Suggestions for Authors

Thank you for the revisions. I can confirm that the comments have now been addressed in the manuscript, and the paper is improved in the areas previously noted.

Author Response

Comment: "Thank you for the revisions. I can confirm that the comments have now been addressed in the manuscript, and the paper is improved in the areas previously noted."

Dear Reviewer,

we like to thank you again for your time and valuable feedback. The article benefited greatly from the feedback. We did not receive any feedback from you regarding the revision. One reviewer had additional comments. The manuscript has been resubmitted with minor additions to clarify certain passages, but does not contain any significant changes. We will perform English editing as soon as the article is accepted in terms of content.